# Thrombotic Risk Detection in Patients with Polycythemia Vera: The Predictive Role of *DNMT3A/TET2/ASXL1* Mutations

**DOI:** 10.3390/cancers12040934

**Published:** 2020-04-10

**Authors:** Adrián Segura-Díaz, Ruth Stuckey, Yanira Florido, Jesús María González-Martín, Juan Francisco López-Rodríguez, Santiago Sánchez-Sosa, Elena González-Pérez, María Nieves Sáez Perdomo, María del Mar Perera, Silvia de la Iglesia, Teresa Molero-Labarta, María Teresa Gómez-Casares, Cristina Bilbao-Sieyro

**Affiliations:** 1Hematology Department, Hospital Universitario de Gran Canaria Dr. Negrín, 35019 Las Palmas de Gran Canaria, Spain; adsegura@hotmail.es (A.S.-D.); rstuckey@funcanis.es (R.S.); floryyana@hotmail.com (Y.F.); juanfra5226@gmail.com (J.F.L.-R.); Jsanchez@funcanis.es (S.S.-S.); gpelena88@gmail.com (E.G.-P.); marysnow@telefonica.net (M.N.S.P.); mperalvm@gobiernodecanarias.org (M.d.M.P.); siglini@gobiernodecanarias.org (S.D.l.I.); tmollab@gobiernodecanarias.org (T.M.-L.); 2Investigation Unit, Hospital Universitario de Gran Canaria Dr. Negrín, 35019 Las Palmas de Gran Canaria, Spain; josu.estadistica@gmail.com; 3Department of Medical Sciences, Universidad de Las Palmas de Gran Canaria, 35016 Las Palmas de Gran Canaria, Spain; 4Morphology Department, Universidad de Las Palmas de Gran Canaria, 35016 Las Palmas de Gran Canaria, Spain

**Keywords:** prognosis, myeloproliferative neoplasms, cardiovascular disease, thrombosis, next-generation sequencing, personalized medicine

## Abstract

The development of thrombotic events is common among patients with polycythemia vera (PV), essential thrombocythemia (ET), and primary myelofibrosis (PMF). We studied the influence of pathogenic mutations frequently associated with myeloid malignancies on thrombotic events using next-generation sequencing (NGS) in an initial cohort of 68 patients with myeloproliferative neoplasms (MPN). As expected, the presence of mutations in *DNMT3A, TET2,* and *ASXL1* (DTA genes) was positively associated with age for the whole cohort (*p* = 0.025, OR: 1.047, 95% CI: 1.006–1.090). Also, while not related with events in the whole cohort, DTA mutations were strongly associated with the development of vascular events in PV patients (*p* = 0.028). To confirm the possible association between the presence of DTA mutation and thrombotic events, we performed a case-control study on 55 age-matched patients with PV (including 12 PV patients from the initial cohort, 25 with event vs. 30 no event). In the age-matched case-control PV cohort, the presence of ≥1 DTA mutation significantly increased the risk of a thrombotic event (OR: 6.333, *p* = 0.0024). Specifically, mutations in *TET2* were associated with thrombotic events in the PV case-control cohort (OR: 3.56, 95% CI: 1.15–11.83, *p* = 0.031). Our results suggest that pathogenic DTA mutations, and particularly *TET2* mutations, may be an independent risk factor for thrombosis in patients with PV. However, the predictive value of *TET2* and DTA mutations in ET and PMF was inconclusive and should be determined in a larger cohort.

## 1. Introduction

The *BCR-ABL1*-negative myeloproliferative neoplasms (MPN) polycythemia vera (PV), essential thrombocythemia (ET), and primary myelofibrosis (PMF) are rare hematological malignancies characterized by the clonal expansion of an abnormal hematopoietic stem cell. Patients with MPN show a highly variable prognosis. Adverse cardiovascular events including thrombosis and hemorrhages are common among patients with MPN, affecting 34–39% of patients with PV, 10–29% of patients with ET, and 7.2–13.2% of patients with PMF [1]. Indeed, adverse cardiovascular events are the major cause of morbidity and mortality in patients with MPN.

Prognostic scores have been developed for each MPN subtype to stratify patients according to the risk of thrombosis and inform their treatment plan. For example, IPSET includes the *JAK2*V617F mutation, a confirmed thrombosis risk factor for ET patients [2]. Other mutations, such as *CALR* type I, have also been associated with an increased risk of thrombosis, while type II *CALR* mutations are associated with a lower incidence of thrombosis compared to *JAK2* mutation [3]. In many cases, thrombotic events are the presenting feature of the disease at diagnosis. However, fatal and non-fatal thrombotic events are frequent during follow-up, even in patients treated with cytoreduction in combination with oral anticoagulants and/or anti-aggregant therapies [4]. Thus, existing algorithms to predict the development of thrombosis in MPN patients require improvements.

Advances in sequencing technology have improved our understanding of the contribution of non-driver mutations as risk factors for overall survival and fibrotic or leukemic transformation. Moreover, the inclusion of somatic mutations in the risk classification of MPN has contributed to the development of more accurate predictive tools [5], such as the mutation-enhanced MIPSS70+ prognostic score for PMF [6] and the online calculator of individualized MPN patient outcomes [7].

Sequencing studies have also demonstrated a direct association between age and the accumulation of somatic mutations, particularly in *DNMT3A, TET2*, and *ASXL1,* often collectively referred to as the DTA genes, as well as *JAK2* [7,8,9,10,11,12]. Indeed, DTA gene mutations have been shown to confer a selective advantage and lead to clonal hematopoiesis (CHIP) [8,9,10,11]. Importantly, individuals with clonal hematopoiesis (even those without a hematologic malignancy) are estimated to have a 1.9-fold higher risk of developing coronary disease and a 4-fold higher risk of myocardial infarction [12].

Despite the recent advances in sequencing techniques and an increased interest in the contribution of non-driver mutations as risk factors, to date, very few studies have investigated the association between mutations in non-driver genes and the development of thrombosis in MPN patients. To address this research need, we aimed to study the impact of gene mutations frequently associated with myeloid malignancies on thrombotic events using next-generation sequencing in a cohort of *BCR-ABL1*-negative MPN patients.

## 2. Results

The initial cohort included a total of 68 patients, consisting of 16 patients with PV, 25 with ET, 16 with PMF, and 11 patients with secondary myelofibrosis (SMF). Patients had an average age of 68 years (range 43–90; median 68 years) and 38.2% were males (patient characteristics are shown in Appendix A). With a median follow-up of 50 months (range 1–200 months), 22 patients developed 25 thrombotic events after diagnosis (32.4% of the MPN cohort), 12 corresponding to arterial thrombosis (48.0% of the total thrombotic episodes) and 13 to venous thrombosis (Table 1). Events were registered in 43.8% of patients with PV (7/16 patients), 40% of patients with ET (10/25 patients), and 18.5% (5/27 patients) with MF.

Among the patients with MPN, 48.5% (33/68 patients) presented a pathogenic non-driver mutation (i.e., in genes other than *JAK2, CALR,* or *MPL;*
Appendix A). The mean number of pathogenic non-driver mutations was 0.63 mutations/patient; 11% of patients presented with more than one pathogenic non-driver mutation. The most frequently mutated genes were *TET2* (32.6%), *ASXL1* (14.0%), and *DNMT3A* (14.0%) (Figure 1A).The presence of mutations in the CHIP-associated DTA genes was 62.5% in PV, 56.3% in PMF, 45.5% in SMF, and 12% in ET (Figure 1B–E).

As expected, we observed a positive association between the presence of DTA mutations and age (*p* = 0.025, OR: 1.047, 95% CI: 1.006–1.090). In the whole series, the presence of DTA mutations or any non-driver mutation were not related with the development of thrombotic events. However, when testing each neoplasia separately, a significant association between the variables of presence of a DTA mutation and vascular event was observed in PV patients (*p* = 0.031, Pearson’s χ^2^). Specifically, 85.7% of patients with PV and an event (6/7 patients) harbored a DTA mutation compared to 33.3% of patients with PV without an event (3/9 patients). When also considering SMF-post PV (6 additional patients, 1 of whom had an event), 87.5% (7/8 patients) with an event had a DTA mutation compared to 42.9% (6/14 patients) without an event (*p* = 0.028, Fisher’s exact test).

The association observed between presence of DTA mutation and thrombotic event in PV patients but not in the whole MPN series could be conditioned by the co-occurrence of a *JAK2* mutation (since almost all patients with PV have *JAK2* mutations compared to 50% in patients with MF, 60% in patients with SMF post-ET, and 48% in patients with ET in our series). To analyze this possibility, we first evaluated if there was an association between the presence of a *JAK2* mutation and thrombotic event in patients with ET, PMF, and SMF post-ET (*n*=44 patients with information available about thrombotic events). Using Pearson’s χ^2^ test we observed an association between the presence of a *JAK2* mutation and thrombotic event in these patients, albeit not statistically significant (*p* = 0.124, Appendix A).We were unable to determine if an association existed between a thrombotic event and the presence of a DTA mutation or any non-driver mutation comparing *JAK2*-mutated versus *JAK2*-wild-type patients due to the low number of events (Appendix A).

To confirm the possible association between the presence of DTA mutation and thrombotic events in PV, we performed a case-control study on 55 age-matched patients (Appendix A) with PV (including 12 of the 16 PV patients from the initial cohort). Of the 55 age-matched patients, 25 experienced a thrombotic event (including 15 events pre-diagnosis or at diagnosis and 16 events post-diagnosis) vs. 30 PV patients without an event (Appendix A). The median follow-up of patients with an event was 108.3 months (P25–P75 44.3–163.0) and 120.6 months (P25–P75 56.2–164.8) for patients without an event.

Following targeted next-generation sequencing (NGS) with the myeloid gene panel, we observed a significant association between the variables of thrombotic event and presence of a DTA mutation for the PV age-matched cohort (*p* = 0.0024, OR: 6.33, 95% CI: 2.02–22.32, Fisher’s exact test; Figure 2): 65.5% of PV patients with an event harbored a DTA mutation vs. 34.5% of PV patients without an event.

Considering mutations in the *TET2* gene in isolation, a positive association existed between thrombotic event and the presence of a pathogenic *TET2* mutation (*p* = 0.031, OR: 3.56, 95% CI: 1.15–11.83, Fisher’s exact test; Figure 3): 65.0% of PV patients with an event harbored a *TET2* mutation vs. 35.0% of PV patients without an event. A significant association was also observed between the presence of any pathogenic non-driver mutation and thrombotic event (*p* = 0.0077, OR: 5.23, 95% CI: 1.63–19.23, Fisher’s exact test; Appendix A). However, when patients with *TET2* mutations were removed and the logistic regression analysis repeated, the association of thrombotic event with *ASXL1/DNMT3A* mutations or any non-driver mutation (other than *TET2*) lost statistical significance (*p* = 0.185 and *p* = 0.487, respectively).

## 3. Discussion

Our study used next-generation sequencing to investigate the impact of pathogenic non-driver mutations commonly associated with myeloid malignancies on thrombotic risk. These mutations were harbored by 48.5% of the MPN patient cohort, in accordance with the previously observed prevalence of 53% for patients with PV or ET [2]. The CHIP-associated DTA genes were the most frequently mutated in the whole MPN cohort, also in accordance with previously published results [7,13]. The frequency of CHIP-associated DTA gene mutations ranged from 12% for patients with ET to 62.5% for patients with PV. In contrast, the estimated frequency is 2–3% in the peripheral blood of healthy individuals, rising to approximately 10% in those aged over 70 years [9].

In patients with ET, PMF, and SMF post-ET, the presence of *JAK2* mutation was marginally associated with thrombotic event. However, since practically all PV patients were mutated in *JAK2,* it is clear that an additional factor must contribute to thrombotic risk in patients with PV. Indeed, our preliminary results reveal that mutations in *DNMT3A, TET2,* and *ASXL1* genes, while not related with thrombotic events in the overall cohort, were strongly associated with events in PV patients. The presence of at least one DTA mutation increased the risk of developing a vascular event by 6-fold in patients with PV. Specifically, mutations in the *TET2* gene in isolation were also associated with thrombosis in the PV case-control cohort. A previous study reported a possible association between *TET2* mutations and thrombosis in ET [13]. However, a 2017 study which investigated the impact of mutations in *JAK2, TET2,* and *ASXL1* genes on thrombotic risk in 587 patients with PV [14] found that *TET2* or *ASXL1* mutations (detected in 18% and 11% of patients, respectively) did not impact arterial nor venous thrombosis [14], although this was not a case-control study.

Our results are in accordance with recent evidence suggesting that the increased cardiovascular risk associated with mutations in genes encoding the epigenetic regulators DNMT3A, TET2, and ASXL1 is due to altered methylation, which causes the increased transcription of pro-inflammatory genes and may trigger atherosclerosis [12,15,16]. For example, a 2012 study showed that loss of TET2 function was associated with atherosclerosis and the increased secretion of pro-inflammatory factors including IL-1β [15]. In our study, *TET2* mutation seems to be a key factor in driving thrombotic events in PV. A larger series would be needed to determine the contribution of *DNMT3A*, *ASXL1,* or other non-driver genes to the risk of developing an adverse cardiovascular event in patients with PV. Moreover, the predictive value of *TET2* and DTA mutations on thrombotic risk in patients with ET and PMF was inconclusive due to the low number of events and the small number of analyzed patients. This association should be determined in larger ET and PMF cohorts and further case-control studies.

Our observations, if confirmed in a larger patient series, could help to identify PV patients at diagnosis who are at higher risk of developing thrombosis. For example, the current risk algorithm for patients with PV, stratified into low risk (<60 years, no history of thrombosis) and high risk (>60 years and/or history of thrombosis) could be modified to include a third very high-risk group (>60 years and/or history of thrombosis with a *TET2* mutation). Such very high-risk patients would benefit from closer monitoring and a modified treatment plan, with the aim of reducing the currently high rates of thrombotic events among MPN patients. For example, preventative therapies with anti-inflammatory agents such as anti-IL-1β might be effective [16]. Randomized clinical trials are required to confirm whether such agents would prevent adverse cardiovascular events in high-risk patients with MPN, although a preliminary study involving patients with CHIP enrolled in the CANTOS study reported that patients with *TET2* mutations responded better to the anti-IL-1β antibody canakinumab than patients without CHIP-associated mutations [17]. It would also be interesting to evaluate if treatment with ruxolitinib, an agent with anti-inflammatory and immunosuppressive properties, can prevent thrombosis in patients with MPN given the reduction of pro-inflammatory cytokines, including IL-6, as a result of its JAK1 inhibitory action [18].

DNA sequencing of patients with MPN using a relatively-inexpensive standardized myeloid panel is not currently part of routine clinical practice. However, such an analysis can provide important prognostic information on vascular risk and overall survival [19].

To conclude, given the continued aging of the global population (with adults aged 65 and over estimated to constitute 16% of the global population by 2050 [20]), the high incidence of MPNs in the over-65 age group, the increasing incidence of DTA mutations with age [7,8,9,10,11] and the association of DTA mutations with both MPN and cardiovascular risk [12], the observation that DTA mutations (especially mutations in *TET2*) may be an independent risk factor for thrombosis in PV is particularly pertinent. It also confirms the importance of a multidisciplinary cardio-onco-hematology approach for the management of patients with MPN.

## 4. Materials and Methods

### 4.1. Ethical Approval

This retrospective non-interventional study was conducted in accordance with the Declaration of Helsinki, and the protocol was approved by our institutional review board (Comité Ético de Investigación Clínica, approval no. ref. 2019-230-1). Informed consent for inclusion was provided by all patients before they participated in the study.

### 4.2. Patients

Patients aged 18 years and above with confirmed diagnosis of PV, ET, or MF according to the 2016-revised WHO criteria were eligible for inclusion. Informed consent was obtained for a total of 68 patients with MPN (initial MPN series)and 55 age-matched PV patients (PV case-control cohort, including 12 of the 16 PV patients from the initial MPN cohort) diagnosed between January 2003 and July 2018 at the Hospital Universitario de Gran Canaria Dr. Negrín, Las Palmas, Spain.

The MPN series included 16 PV, 25 ET, 16 PMF, and 11 SMF patients. Patients with MF were taken consecutively; the PV and ET patients were retrospectively chosen from a previous research project (with different objectives to this study) and the status of the patients’ thrombotic events was unknown at the time of their selection. 

For the case-control cohort, all patients diagnosed at our hospital with a confirmed diagnosis of PV and thrombotic events in their patient history were taken as “cases” (*n* = 25). Control patients (with a diagnosis of PV but no thrombotic event) were taken consecutively and age-matched with the cases (*n* = 30). Control patients that did not age-match with a control were removed.

See Appendix A for patient characteristics of the initial MPN cohort and PV case-control cohort, respectively.

### 4.3. Next-Generation Sequencing (NGS)

NGS was performed on 200 ng genomic DNA extracted from peripheral blood at diagnosis. Sequencing was performed using a MiSeq (Illumina, San Diego, CA, USA) platform with the targeted 30-gene panel Myeloid Tumor Solution™ (SOPHiA Genetics, Saint-Sulpice, Switzerland). Variants were identified using the analysis software SOPHiA DDM (4.8.1.3). Only variants with an allelic frequency (VAF) ≥ 2% (established as cutoff for mutations associated with CHIP in peripheral blood [6]), a described population frequency (MAF) < 1%, and an annotated pathogenic effect (or probability >90% of being pathogenic) were included, with pathogenicity determined according to ClinVar (www.ncbi.nlm.nih.gov/clinvar/), COSMIC (https://cancer.sanger.ac.uk/cosmic), VarSome (https://varsome.com/), ENSEMBL (https://www.ensembl.org/index.html), and IARC TP53 databases (http://p53.iarc.fr/TP53GeneVariations.aspx), and published studies. 

### 4.4. Statistical Analysis

Chi-squared tests and logistic regression analyses were performed to analyze the impact of the independent variables on thrombotic risk. Odd-ratios (OR) with correspondent *p* values and confidence intervals (CI) were estimated. All *p* values <0.05 were considered statistically significant. Analyses were performed using statistical software R Core Team 2020, version 3.6.2.

## 5. Conclusions

Our study investigated the association between pathogenic mutations in non-driver genes and the development of thrombosis, the major cause of morbidity and mortality in patients with *BCR-ABL1*-negative MPN. Pathogenic DTA mutations increased thrombotic risk by 6-fold in PV patients, although the association with patients with MF or ET was inconclusive. Mutations in *TET2* seemed to be a key factor in driving thrombotic events in PV. Our results are in accordance with recent evidence linking *TET2* mutations with atherosclerosis and increased cardiovascular risk. Our observations could be applied to improve current risk algorithms for the more accurate identification of PV patients at diagnosis who are at higher risk of developing thrombosis.

## Figures and Tables

**Figure 1 cancers-12-00934-f001:**
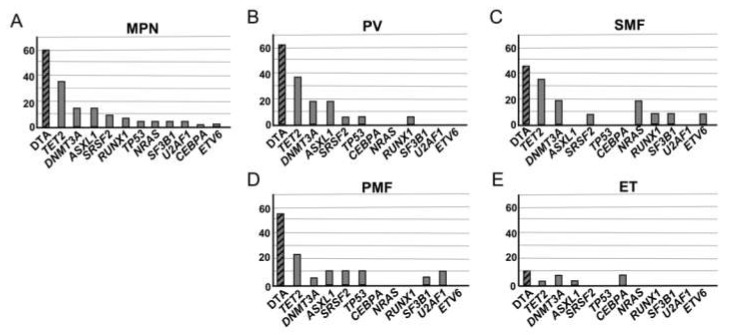
Most frequent non-driver gene mutations. (**A**) For the whole patient cohort (*n* = 68), and for each neoplasia separately, (**B**) polycythemia vera (PV) (*n* = 16), (**C**) secondary myelofibrosis (SMF) (*n* = 11), (**D**) primary myelofibrosis (PMF) (*n* = 16), and (**E**) essential thrombocythemia (ET) (*n* = 25). DTA genes (*DNMT3A, TET2, ASXL1*) were the most frequently mutated in both the whole cohort and individual neoplasias. Hatched bars represent the sum of mutations in the DTA genes.

**Figure 2 cancers-12-00934-f002:**
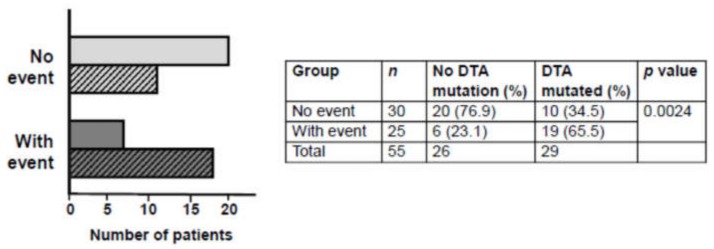
Frequency of DTA mutations in the no thrombotic event and with thrombotic event groups of PV patients. A significant association was observed between event and the presence of DTA mutation. Open bars represent no DTA mutation, hatched bars represent a DTA mutation. DTA: *DNMT3A, TET2, ASXL1*. Odds ratio 4.68 (95% confidence interval 1.493–14.644).

**Figure 3 cancers-12-00934-f003:**
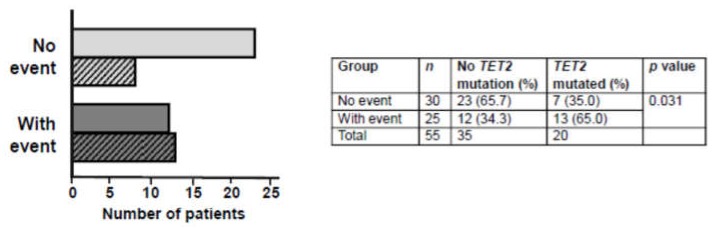
Frequency of *TET2* mutations in the no thrombotic event and with thrombotic event groups of PV patients. Open bars represent no *TET2* mutation, hatched bars represent *TET2* mutation. Odds ratio 3.11 (95% confidence interval 1.012–9.582).

**Table 1 cancers-12-00934-t001:** Arterial and venous thrombotic events registered in the myeloproliferative neoplasms (MPN) cohort at or after diagnosis.

	MPN	First Event	Second Event
1	PV	IS	
2	PV	RVO	
3	PV	MI	
4	PV	Budd–Chiari	
5	PV	MI	
6	PV	DVT	TIA
7	PV	DVT and PE	
8	ET	PVT	
9	ET	MI	
10	ET	PE	
11	ET	Budd–Chiari	
12	ET	Acute carotid stroke	
13	ET	TIA	
14	ET	DVT	
15	ET	IS	MI
16	ET	MI	
17	ET	SVT	
18	PMF	PVT	
19	PMF	PVT	
20	PMF	MI	
21	PMF	PE	
22	SMF (post-PV)	PE	PVT

IS: ischemic stroke, RVO: retinal vein occlusion, MI: myocardial infarction, DVT: deep vein thrombosis, PE: pulmonary embolism, TIA: transient ischemic attack, PVT: portal vein thrombosis, SVT: superficial vein thrombosis.

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
