# Peer review of "Thrombotic Risk Detection in Patients with Polycythemia Vera: The Predictive Role of DNMT3A/TET2/ASXL1 Mutations"

_cancers, 2020, doi:10.3390/cancers12040934_

Round 1

Reviewer 1 Report

The aim of this paper was to evaluate the prognostic significance of DTA genes mutations in assessing possible thrombotic events in patients with myeloid malignancies. Using next-generation-sequencing the authors confirmed the well-known fact of increasing gene mutations with age and also showed association of DTA mutations with the development of vascular events in PV patients. Additionally, in the age-matched case control PV cohort it was shown that DTA mutations (and particularly TET2 mutations) were associated with thrombotic events.

Based on this, DTA (TET2) mutations have been proposed by the authors as independent risk factors for thrombosis development in PV.

The importance of the thrombotic complications problem in MPN and its lack of knowledge determine the relevance of this study. Therefore, the identification of new independent risk factors for thrombosis in MPN (and particularly in PV) patients is not only of theoretical but also of practical importance. The study is not distinguished by absolute novelty - the authors once again confirmed the well-known fact of an increase in DTA genes mutations with age. But at the same time, original data were obtained on the possibility of using DTA (TET2) mutations as independent risk factors for thrombosis in patients with PV. In addition to the existing stratification of risk factors, the authors propose to differentiate third very high-risk group (>60 years and/or history of thrombosis with a TET2 mutation).

In general, the article seems to be clearly structured, logical, consistent, the authors demonstrated good knowledge of the current state of the problem; however, some points require clarification and adjustment:

  1. Section2 Results, lines 82-83: it seems not quite correct indication of % ratio of the patients with thrombotic events, because in PV patients 6 ones have had thrombotic events (1 of them – twice), and this is 6/16 (not 7/16)(37.5%), the same doubtful indication is given for patients with PMF.
  2. 2 and the tables nearby (the same for Fig. 3) duplicate each other. Abbreviations should be defined in parentheses the first time they appear in the text (TACE? – fig.2 and fig. 3)
  3. Section 4.2 states that informed consent was obtained for a total of 68 patients with MPN, and what about the informed consent for age-matched PV cohort (excluding 12 PV patients from initial cohort)?

Author Response

We thank the reviewer for their helpful comments.

  1. Section 2 Results, lines 82-83: it seems not quite correct indication of % ratio of the patients with thrombotic events, because in PV patients 6 ones have had thrombotic events (1 of them – twice), and this is 6/16 (not 7/16)(37.5%), the same doubtful indication is given for patients with PMF.

The number of PV patients with events should be 7/16 as originally indicated in the text (this has been corrected in Table 1); but 5/16 patients for MF. The results have been corrected in the text.

  1. 2 and the tables nearby (the same for Fig. 3) duplicate each other. Abbreviations should be defined in parentheses the first time they appear in the text (TACE? – fig.2 and fig. 3)

Figure 2 contains a bar graph and its respective contingency table. We chose to include the bar graph since it gives a simple visual representation of the data, clearly showing the association between event and DTA mutation in the case of Figure 2. We also included the contingency table as, although as you point out, the table duplicates the information represented in the bar graph (the number of patients with/without events and with/without DTA mutation), it also contains the total number of patients in each group, the percentages, and the p-value as based on Pearson’s chi-squared test. Therefore, although we agree that there is a certain degree of duplication in this figure (and similarly Figure 3), we believe the inclusion of the contingency table both helps to summarize the relationship observed and provides additional information to the reader. However, if the reviewer thought it necessary, we would be happy to remove the contingency tables from Figures 2 and 3 and include these as supplementary tables.

We have removed all mention of TACE from the revised manuscript; this was an oversight.

  1. Section 4.2 states that informed consent was obtained for a total of 68 patients with MPN, and what about the informed consent for age-matched PV cohort (excluding 12 PV patients from initial cohort)?

Thank you for bringing this oversight to our attention; informed consent was also obtained from patients in the age-matched PV cohort. The text in Section 4.2 has been modified accordingly.

Reviewer 2 Report

In this paper the authors report on the correlation in a cohort of PV patients between the presence of non-driver mutations and the occurrence of thrombotic events. The identified that mutations in one of the ASXL1, DNMT3A and/or TET2 genes is positively correlated to an increased thrombotic risk.

The results sound interesting but the study is limited due to the absence of a clear definition of the cohorts used.

Major points

  • in the text it is stated that 7 patients with PV had thrombotic events but in Table 1 only 6 patients with PV are reported. Where is the good result ?
  • In a first part the author report on a chart of 68 MPN patients among which 16 had PV. It is not described how these patients were chosen. Why these patients and not others ? It is mandatory to show that no bias could derive from the choice of the patients studied. Then in a second part the authors describe a "case-control" study on 55 PV patients that included 12 (75%) of the patients from the first part. I do not understand how this is a case-control study. Who are the cases and who are the controls ? Once again how did they choose the patients ? 

For this study to be more robust the patients should be taken prospectively or even retrospectively but consecutively with no choice. Was it the case ?

Minor points:

  • on Supplementary Table 1 I do not understand the meaning of the "History of thrombosis". If this means occurrence of thrombosis before the diagnosis then this should be stated more clearly.
  • in the supplementary tables, it would be more adequate to present the mutations at the protein level instead of the cDNA level (or even better to present both). Indeed, the protein nomenclature for mutations is the most widely used and readers will take more advantage of this information. As an example the DNMT3A p.R882H mutation is well known while rather nobody knows about c.2645G>A.

Author Response

Major points

  • in the text it is stated that 7 patients with PV had thrombotic events but in Table 1 only 6 patients with PV are reported. Where is the good result ?

Thank you for bringing this oversight to our attention. The number of PV patients with events is 7/16 as originally indicated in the text (we have updated Table 1 accordingly).

  • In a first part the author report on a chart of 68 MPN patients among which 16 had PV. It is not described how these patients were chosen. Why these patients and not others ? It is mandatory to show that no bias could derive from the choice of the patients studied.

The first MPN patient series came from a retrospective selection of MPN patients diagnosed at our hospital for NGS analysis for an earlier project of investigation. In the case of patients with MF, the patients were taken consecutively. However, PV and ET patients were selected with different objectives to those for this study and the status of thrombotic events was unknown at the time of their selection. The % of JAK2 mutations among this cohort is as described in the published literature (50% in patients with MF, and 48% in patients with ET), as is the % of triple-negative MPN patients (20% of ET patients). This series is described with more detail in the methods section (lines 224-227).

            We can understand the reviewer’s concern whether bias was introduced into the MPN cohort. We do not believe this to be the case, particularly as the thrombotic status of these patients was not known at the time of their selection. However, if the reviewer still has concerns in relation to the MPN cohort, we would be willing to remove the first set of results stemming from this overall MPN cohort (lines 79-122, with removal of Table 1 and Figure 1). Thus, the results section would start at line 123. This removal would not change the key results presented in our manuscript or the key conclusions made, namely that DTA mutations, and particularly TET2 mutations, have a possible predictive role for thrombotic risk in PV patients.

Then in a second part the authors describe a "case-control" study on 55 PV patients that included 12 (75%) of the patients from the first part. I do not understand how this is a case-control study. Who are the cases and who are the controls ? Once again how did they choose the patients ? 

For this study to be more robust the patients should be taken prospectively or even retrospectively but consecutively with no choice. Was it the case ?

Of the patients diagnosed with PV at our hospital, we retrospectively performed NGS on all of those with reported thrombotic events (n=25 “cases”). Therefore, these PV patients were not chosen and thus there was no possible bias. Control patients (with a diagnosis of PV but no thrombotic event) were taken consecutively and age-matched with the cases (n=30). Control patients that did not age-match with a control were removed from the database. As the control patients were taken consecutively, they were neither chosen nor possible bias introduced. This is described with more detail in the methods section (lines 228-231).

Minor points:

  • on Supplementary Table 1 I do not understand the meaning of the "History of thrombosis". If this means occurrence of thrombosis before the diagnosis then this should be stated more clearly.

The heading of the “History of thrombosis” column has been changed to “Occurrence of thrombosis before diagnosis”, in Supplementary Tables 1 and 5 (old Supplementary Table 3), to make this clearer.

  • in the supplementary tables, it would be more adequate to present the mutations at the protein level instead of the cDNA level (or even better to present both). Indeed, the protein nomenclature for mutations is the most widely used and readers will take more advantage of this information. As an example the DNMT3A p.R882H mutation is well known while rather nobody knows about c.2645G>A.

As suggested by the reviewer, we have also included the mutations at the protein level in Supplementary Tables 1 and 5 (old Supplementary Table 3).

Reviewer 3 Report

Authors describe a possible predictive role of DNMT3A/TET2/ASXL1 mutations for thrombotic risk in PV patients. 

Major comments:

According to the new WHO classification, it is more appropriate to speak about bcr\all negative and not Ph- MPNs;

It is difficult to understand, at a first reading, the difference between data in figure 2 and in figure 3 part B. To simplify one of the two might be removed;

In the discussion, when discussing anti-inflammatory therapies (rows 174-180), also ruxolitinib should be taken into consideration (ref. Elli et al. Front Oncol. 2019 Nov 7;9:1186. doi: 10.3389/fonc.2019.01186. eCollection 2019. Review)

A possible predictive role is reported in PV and not in TE or MF (maybe in PPV-MF): this may be due to the fact that almost all PV have JAK2 mutations, while in TE and MF JAK2 mutated patients are 55-60% "only". This should be discussed and possibly analyzed, looking at JAK2 mutated  vs. JAK2 wild type patients and presence/absence of TET2 or DTA mutations.

Although reported data are interesting, these may be misleading due to the small number of analyzed patients.

MInor comments:

DTA and TACE abbreviations are never explained in the text, even in their first occurrence;

row 26: a space between events and in is missing;

row 146: a space between PV. and In is missing;

row 152: a space between thrombosis and in is missing;

row 153: spaces between ref.7 and However and between a and 2017 are missing;

row 164: a space between events and in is missing;

row 183: reference is missing (for example Front Oncol. 2019 Apr 26;9:321. doi: 10.3389/fonc.2019.00321. eCollection 2019. Review).

row 222: a space between thrombotic and risk is missing;

row 223: a space between in and TET2 is missing;

Author Response

Major comments:

According to the new WHO classification, it is more appropriate to speak about bcr\all negative and not Ph- MPNs;

We have changed the nomenclature of Ph- MPN for BCR-ABL1-negative MPN throughout the manuscript as suggested by the reviewer.

It is difficult to understand, at a first reading, the difference between data in figure 2 and in figure 3 part B. To simplify one of the two might be removed;

As suggested by the reviewer, Figure 3B has been removed (now new Supplementary Figure 2) to make these results easier to understand.

In the discussion, when discussing anti-inflammatory therapies (rows 174-180), also ruxolitinib should be taken into consideration (ref. Elli et al. Front Oncol. 2019 Nov 7;9:1186. doi: 10.3389/fonc.2019.01186. eCollection 2019. Review)

We have added an extra phrase into the discussion to mention the anti-inflammatory properties of ruxolitinib and included the suggested reference as new reference 18.

A possible predictive role is reported in PV and not in TE or MF (maybe in PPV-MF): this may be due to the fact that almost all PV have JAK2 mutations, while in TE and MF JAK2 mutated patients are 55-60% "only". This should be discussed and possibly analyzed, looking at JAK2 mutated  vs. JAK2 wild type patients and presence/absence of TET2 or DTA mutations.

The impact of JAK2 mutations on thrombotic events was not discussed in the original manuscript since, as stated by the reviewer, almost all PV have JAK2 mutations, thus there must be a further contributory factor. In the revised manuscript we have included a statistical test (new Supplementary Table 3) to analyze the role of JAK2 on events in the MPN total series (removing the 16 PV and 6 SMF post-PV), for which we observed an association but with a p-value of 0.124 (lines 111-119).

We also evaluated the association with presence of DTA mutations and any non-driver mutation in JAK2-mutated vs JAK2 wild-type patients (new Supplementary Table 4); however the number of events was very low, thus it was impossible to form any conclusions in the TE and MF patients (lines 119-121).

This result is also mentioned in the discussion, lines 162-164.

Although reported data are interesting, these may be misleading due to the small number of analyzed patients.

As mentioned in the original manuscript, our findings in PV are preliminary and should be confirmed in a larger series. Larger cohorts of TE and MF patients are also required in order to determine the predictive value of DTA and/or TET2 mutations in these patients. This is discussed in lines 182-185.

MInor comments:

DTA and TACE abbreviations are never explained in the text, even in their first occurrence;

In the original manuscript DTA was defined in the abstract (line 24), again in the introduction (line 67) and in the figure legend of Figures 1 and 2, and Supplementary Tables 1 and 3.

We have removed all mention of TACE from the revised manuscript.

row 26: a space between events and in is missing;

row 146: a space between PV. and In is missing;

row 152: a space between thrombosis and in is missing;

row 153: spaces between ref.7 and However and between a and 2017 are missing;

row 164: a space between events and in is missing;

row 222: a space between thrombotic and risk is missing;

row 223: a space between in and TET2 is missing;

We apologize for the spacing issue; it appears to have been caused by different Word versions. We have included the spaces as specified by the reviewer and double-checked the manuscript thoroughly for correct spacing throughout.

row 183: reference is missing (for example Front Oncol. 2019 Apr 26;9:321. doi: 10.3389/fonc.2019.00321. eCollection 2019. Review).

Thank you for bringing this interesting review to our attention. We have included this in the discussion as new ref. 18.

Round 2

Reviewer 2 Report

In this new version the paper is correct for publication

Reviewer 3 Report

In the new version, all raised points have been amended.

No other changes are required.